# Structure of SRSF1 RRM1 bound to RNA reveals an unexpected bimodal mode of interaction and explains its involvement in SMN1 exon7 splicing

Antoine Cléry[1,2 ✉], Miroslav Krepl[3], Cristina K. X. Nguyen[1], Ahmed Moursy[1], Hadi Jorjani[4], Maria Katsantoni[4], Michal Okoniewski[5], Nitish Mittal[4], Mihaela Zavolan[4], Jiri Sponer[3] & Frédéric H.-T. Allain[1 ✉]

The human prototypical SR protein SRSF1 is an oncoprotein that contains two RRMs and plays a pivotal role in RNA metabolism. We determined the structure of the RRM1 bound to RNA and found that the domain binds preferentially to a CN motif (N is for any nucleotide). Based on this solution structure, we engineered a protein containing a single glutamate to asparagine mutation (E87N), which gains the ability to bind to uridines and thereby activates SMN exon7 inclusion, a strategy that is used to cure spinal muscular atrophy. Finally, we revealed that the flexible inter-RRM linker of SRSF1 allows RRM1 to bind RNA on both sides of RRM2 binding site. Besides revealing an unexpected bimodal mode of interaction of SRSF1 with RNA, which will be of interest to design new therapeutic strategies, this study brings a new perspective on the mode of action of SRSF1 in cells.

[1] Department of Biology, Institute of Biochemistry, ETH Zurich, Zurich, Switzerland. [2] Biomolecular NMR Spectroscopy Platform, ETH Zurich, Zurich, Switzerland. [3] Institute of Biophysics of the Czech Academy of Sciences, Kralovopolska 135, 612 65 Brno, Czech Republic. [4] Computational and Systems Biology, Biozentrum, University of Basel, Basel, Switzerland. [5] Scientific IT Services, ETH Zurich, Zurich, Switzerland. ✉email: aclery@mol.biol.ethz.ch; allain@mol.biol.ethz.ch

RSF1 is the first identified member of the SR protein family, which is characterized by the presence of a domain enriched in arginine and serine residues[1–3]. In addition to being an oncoprotein, SRSF1 was shown to be involved in several steps of RNA metabolism, including alternative splicing regulation, mRNA transcription, stability and nuclear export, translation, nonsense-mediated mRNA decay, sumoylation, and miRNA processing[4]. An important example of SRSF1 involvement in alternative splicing regulation is linked to spinal muscular atrophy (SMA). SMA is characterized by the progressive degeneration of spinal motoneurons, subsequent muscle weakness, and atrophy. The disease is caused by deletion or mutation within survival of motor neuron 1 (SMN1) gene. The SMN2 gene, a nearly identical copy of SMN1 (>99.9% sequence identity), fails to prevent SMA mostly due to a critical C-to-U mutation at position +6 of exon7[5,6]. This nucleotide change does not alter an amino acid but rather promotes the recruitment of two antagonist splicing regulators. It was proposed that the presence of a cytosine induces the binding of the splicing activator SRSF1 on the SMN1 exon7 ESE1 (Exonic Splicing Enhancer 1), which activates exon7 inclusion[7]. In SMN2 exon7, the corresponding uridine was reported to prevent the recruitment of SRSF1 and instead initiate the binding of the splicing repressor hnRNP A1, which primarily promotes the skipping of the exon[8]. As a consequence, the majority of SMN2 transcripts lacks exon7, which leads to a C-terminal truncated SMN protein that is unstable and gets rapidly degraded[9]. In addition to the recruitment of SRSF1 or hnRNP A1, additional splicing factors were shown to be involved in the splicing regulation of the exon7 and the structure adopted by the pre-mRNA also plays a role in this regulation[10]. Despite the fact that SMN2 exon7 splicing is altered, the gene still produces some full-length transcripts (~10%), and hence some SMN proteins[9]. However, this is not sufficient to compensate for a loss of SMN1 expression. Consequently, increasing the amount of functional SMN protein by restoring a SMN1-like splicing pattern from the SMN2 gene was one of the most promising approaches to treat and possibly cure SMA. Using this strategy, an antisense oligonucleotide (ASO) that prevents the binding of hnRNP A1 to the intronic splicing silencer ISS-N1 located in intron7 was designed and produced as a therapeutic molecule[11,12]. This drug (SPINRAZA®) has recently been approved by the Food and Drug Administration as the first treatment against SMA[13].

Like all members of the SR protein family, SRSF1 is characterized by the presence of a C-terminal RS domain enriched in arginine and serine residues, which is primarily involved in protein–protein interactions[14]. The cellular localization of this protein as well as its interaction with RNA and other proteins depends on the phosphorylation state of this domain[14,15]. A structural model of the N-terminal part of the SRSF1 RS domain (RS1) was determined and showed that the phosphorylation of this peptide induces a shift from a fully disordered state to a partially rigidified arch-like structure[16]. In addition to the RS domain, SRSF1 contains two RNA recognition motifs (RRM1 and RRM2) that are responsible for its specific interaction with RNA[14]. Structures of the pseudo-RRM (RRM2) in its free form[17,18] and bound to the SRPK1 kinase[19] and to RNA[20] were determined. The structure of the protein–RNA complex showed that the domain recognizes a GGA motif using a non-canonical mode of interaction centered on the α1-helix of the RRM[20]. A more recent study also suggested that the unphosphorylated N-terminal part of the RS domain (RS1) could interact in *trans* with SRSF1 RRM2[17] and arginines contained in the inter-RRM linker were shown to be involved in SRSF1 interaction with TAP mRNA export factor[18]. However, no structural data are available for SRSF1 RRM1 so far. Nevertheless, the N-terminal extremity of this domain was shown to play an important role in pre-mRNA

splicing[21] and a SELEX experiment indicated that the protein could bind specifically to RNA[22]. A GGAGA-binding consensus motif was identified from CLIP-seq experiment performed with this protein[23] and was consistent with previous investigations[22,24–31]. However, this identified targeted sequence could not explain the specific interaction of SRSF1 with the cytosine found at position +6 of SMN1 exon7.

Here we investigated the mode of RNA recognition of SRSF1 RRM1 and found that it binds preferentially to cytosines and recognizes the CA motif found in SMN2 exon7. By solving the solution structure of this complex, we could engineer a version of SRSF1 RRM1 containing a single glutamate to asparagine substitution (E87N) that binds to uridines. Remarkably, this protein was then able to activate the splicing of SMN2 exon7, in which the cytosine at position +6 is replaced by a uridine. Finally, we also investigated the mode of interaction with RNA of both SRSF1 RRMs linked by their natural linker and showed that RRM1 can bind RNA upstream or downstream of RRM2-binding site due to its flexible inter-RRM linker. This result explains the difficulties of traditional approaches to identify a consensus binding sequence for this protein and reveals almost 30 years after its discovery an unexpected bimodal mode of RNA recognition.

## Results

**SRSF1 RRM1 binds preferentially to cytosines.** SRSF1 contains an N-terminal canonical RRM (RRM1) followed by a non-canonical RRM (RRM2) called pseudo-RRM (Fig. 1A). The solution structure of RRM2 bound to RNA revealed that the domain interacts specifically with a GGA motif using an unusual and conserved mode of RNA recognition that involves primarily conserved residues of the SWQDLKD motif located in the α1-helix of the domain[20]. However, this result could not explain the specific interaction of SRSF1 observed with the cytosine located at position +6 of SMN1 exon7[7]. As a consensus sequence enriched in CG dinucleotides was previously obtained by SELEX with SRSF1 RRM1 (5′-ACGCGCA-3′)[22], we investigated whether SRSF1 RRM1 could be responsible for the recognition of this +6 cytosine. Using nuclear magnetic resonance (NMR), we tested RRM1 binding to polyA, polyC, polyG, and polyU sequences (Fig. S1A). Surprisingly, chemical-shift perturbations were only observed in the presence of polyC showing a strong preference of the human SRSF1 RRM1 for cytosines. Interestingly, we obtained the same trend with the RRM1 of the human SRSF4, SRSF5, SRSF6, SRSF9, and *Drosophila* B52 (Fig. S2) suggesting that this preferential binding to cytosines may be conserved in other RRM1 of SR proteins containing two RRMs. This result already indicated that RRM1 could indeed be responsible for the specific interaction of SRSF1 with SMN1 exon7 +6 cytosine. Indeed, we could see that RRM1 interacts with the SMN1 exon7 ESE1 (UU<u>C</u>AGA) but not with the corresponding sequence in SMN2 exon7 (UU<u>U</u>AGA) (Fig. S1B). As the sequence targeted by SRSF1 in SMN1 exon7 contained a CA dinucleotide instead of the CG-rich motif selected with SELEX, we investigated the importance of the nucleotide at the second position by titrating SRSF1 RRM1 with NNCGNN, NNCCNN, NNCTNN, or NNCANN single-stranded DNAs (ssDNAs; N is for A, C, G, or T). Random nucleotides were present at positions 1, 2, 5, and 6 to prevent any interference of the surrounding nucleotides on the binding efficiency of the domain to the nucleotide at position 4. In good agreement with the results obtained with SELEX[22], ssDNA containing the CG dinucleotide induced the largest chemical-shift perturbations overall indicating a slightly higher affinity of the domain for this motif (Fig. S3). Nevertheless, the pattern of chemical-shift perturbations was similar in the presence of CA,

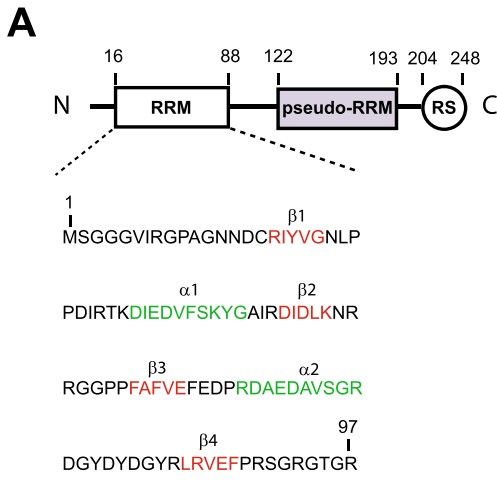

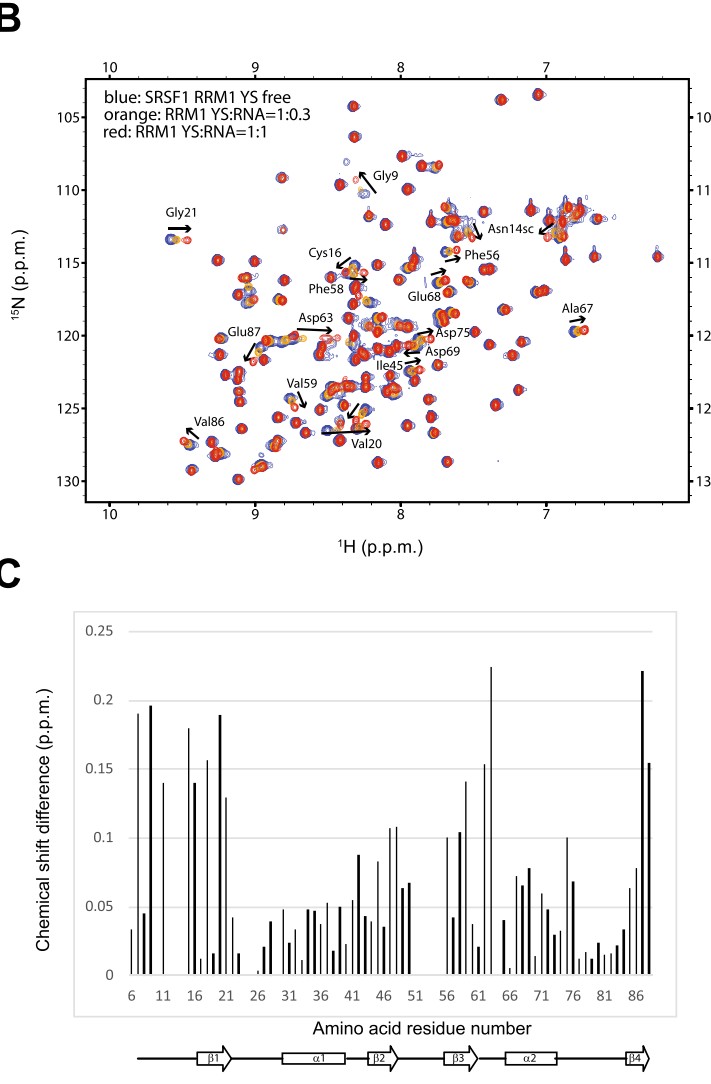

**Fig. 1 SRSF1 RRM1 interaction with the AACAAA RNA. A** The sequence of recombinant SRSF1 protein used in this study is shown. Amino acid numbering is according to the PDB sequence. Amino acids involved in the formation of β-strands and α-helices are colored in red and green, respectively. **B** Superimposition of $^1$H-$^{15}$N HSQC spectra obtained with $^{15}$N-labeled SRSF1 RRM1 and increasing amount of unlabeled 5'-AACAAA-3' RNA. The titration was performed at 40 °C (313 K), in the NMR buffer. The peaks corresponding to the free and RNA-bound states (RNA:protein ratios of 0.3:1 and 1:1) are colored blue, orange, and red, respectively. The highest chemical-shift perturbations observed upon RNA binding are indicated by black arrows. **C** Representation of the combined chemical-shift perturbations ($\Delta\delta = [(\delta\,HN)^2 + (\delta N/6.51)^2]^{1/2}$) of SRSF1 RRM1 amides upon binding to 5'-AACAAA-3' RNA, as a function of RRM1 amino acid sequence. Secondary-structure elements of the protein domain are displayed at the bottom of the graph.

CC, or CT indicating that the second nucleotide interacts with the same pocket of RRM1 independently of its identity. In summary, our NMR investigation revealed the minimal motif bound by SRSF1 RRM1 to be CN, which is on one hand in agreement with the previous SELEX consensus sequence as this motif is present three times and on the other hand more precise as only two nucleotides are really bound to the RRM. Surprisingly, chemical-shift perturbations of backbone amides located in the N-terminal part of the RRM were only observed in the presence of the CA dinucleotide found in SMN2 exon7 ESE1 (Fig. S3B) and was confirmed by the comparison of chemical-shift perturbations observed between AACAAA and AACGAA RNAs (Figs. 1B, C and S1C for Gly9 and Asn14). To better understand the mode of interaction of SRSF1 RRM1 with this CA motif present in SMN1 exon7, we then determined the structure of the domain bound to the 5'-AACAAA-3' RNA.

**Structure of SRSF1 RRM1 bound to RNA**. We noticed that SRSF1 RRM1 tends to aggregate at the high protein concentration required for structure determination and therefore had to mutate two solvent accessible tyrosine residues located in α-helices of RRM1 (Tyr37 and Tyr72: RRM1 YS) to serine. Importantly, these mutations did not change the mode of interaction of the protein with RNA (Fig. S4A, B). We calculated the structure of this complex using 1537 nuclear Overhauser effect (NOE)-derived distance restraints, including 33 intermolecular NOE-derived distance restraints between protein and RNA (Supplementary Table 1). We obtained an ensemble of 20 structures (Fig. 2A) with an r.m.s. deviation of 1.71 Å for all the heavy atoms (Supplementary Table 1).

The RRM in complex with RNA adopts a canonical $\beta_1\alpha_1\beta_2\beta_3\alpha_2\beta_4$ fold that is unchanged compared to the apo form of the protein (PDB number 1X4A). The structure validates the recognition of a CA motif by the RRM using the canonical β-sheet surface. All nucleotides are in the C2'-endo sugar conformation. In good agreement with the chemical-shift perturbations observed in the N-terminal part of RRM1 (Fig. S3B), inter-molecular NOEs were observed between the H2 of $A_4$ and side chain protons of Ile7 and Arg8 (Fig. S5). However, these restraints were not sufficient to precisely describe the interface between the N-terminal part and the RNA. Therefore, we decided to perform molecular dynamics (MD) simulations of this protein–RNA complex and found that Arg8 side chain could interact with the phosphate located between $A_4$ and $A_5$ (Fig. 2C). In addition, the simulations revealed that the carbonyl of Asn14 side chain could form an H bond with the amino group of $A_4$ participating directly in the recognition of the adenine (Fig. 2C). In good agreement with this result, chemical-shift perturbations of protons of the Asn14 side chain amino group were observed upon binding to CA- and not CG-containing RNA (Fig. S1C). Moreover, the substitution of Asn14 by an Alanine strongly reduced the chemical-shift perturbations observed upon RNA binding (Fig. S6A). Similar result was observed with the Arg8 to Alanine mutation (Fig. S6A) confirming that these two residues are important for the N-terminal interaction with $A_4$. Lastly, the simulations have shown that, even though the RNA interactions with Arg8 and Asn14 are quite stable once established, the N-terminus is also very dynamic and samples a great variety of alternative conformations. Lack of a single dominant binding mode in this part of the complex could explain why only two intermolecular NOEs could be detected.

All the intermolecular interactions found in the solution structure were also stable in MD simulations: the stacking of $C_3$ and $A_4$ on Tyr19 and Phe58 rings, respectively, the H bonds between the Glu87 side chain and $C_3$ amino group, Tyr19

hydroxyl with $C_3$ phosphate, Arg17 side chain with $A_4$ N1, and Lys48 side chain with $A_4$ N3 and Asp46 side chain (Fig. 2B). In addition, MD simulations revealed that in some cases a water-mediated interaction between Lys48 and $A_4$ N3 allowed the Lys48 side chain to form an additional H bond with the 2'-OH of $A_4$ ribose (Fig. 2C) suggesting that RRM1 preferentially binds RNA over DNA molecules. In good agreement with the structure and the simulations, mutations to alanine of all residues involved in these interactions with RNA induced a significant decrease of most chemical-shift perturbations observed upon RNA binding at equivalent stoichiometry (Fig. S6), which indicates a decrease in affinity of all these protein mutants for the 5'-AACAAA-3' RNA. In addition, we performed isothermal titration calorimetry (ITC) titrations of the SRSF1 RRM1 YS with AACAAA and AACGAA RNAs and obtained similar $K_d$ values (21 and 25 μM, respectively; Fig. S4C). In good agreement with the NMR data (Fig. 3A), a binding was observed with the polyC RNA and not polyU. The affinity was slightly higher than with the AACAAA and AACGAA RNAs ($K_d$ of 11 μM instead of 21–25 μM), which is most likely due to an avidity effect.

As mentioned above, only the presence of the CA-binding motif induced chemical-shift perturbations in the N-terminal residues of RRM1, although the domain has a similar affinity for both CG- and CA-containing RNAs (Fig. S4C). To better characterize this effect, we performed MD simulations of SRSF1 RRM1 bound to the AACGAA RNA and confirmed the absence of interaction between $G_4$ and the N-terminal part of the domain. Instead, our simulations indicated that it preferentially formed direct hydrogen bonds with the Arg17 and Asp46 side chains (Fig. 2D). Furthermore, the base was also connected to Asp44 and Glu60 side chains through a series of water bridges. The presence of these interactions prevents the access of the N-terminal residues to $G_4$. Indeed, the amino group of $A_4$ is replaced by a carbonyl in $G_4$, which cannot form an hydrogen bond with Asn14 as it is already interacting with the Arg17 located in the β1 strand. These MD data provide a rational for the lack of chemical-shift perturbations observed with the N-terminus of RRM1 upon binding to the AACGAA RNA. In addition, it also explains the similar binding affinity of the domain for both RNAs as the same number of hydrogen bonds are involved in the recognition of $A_4$ and $G_4$. In conclusion, except for the above mentioned differences, both the CA and CG motifs use the same interface to interact with SRSF1 RRM1.

**Engineering of an SRSF1 RRM1 protein that binds to uridines and activates splicing of SMN2 exon7**. SRSF1 was shown to be an activator of exon7 inclusion in SMN1 and not SMN2 due to the C to U nucleotide difference at position +6 of the exon[7]. Here we showed that RRM1 is responsible for the recognition of the cytosine and the solution structure explains well this specific interaction (Fig. 2). We, then wondered whether it would be possible, based on the structure, to engineer a version of SRSF1 RRM1 that could interact with uridines in addition to cytosines. This could then allow the binding of SRSF1 to SMN2 exon7 that contains a uridine at position +6 of the exon and possibly induce its splicing. As the SMN2 gene is the only source of production of SMN proteins in cells of SMA patients, such a protein variant may be an interesting therapeutic strategy to increase the cellular level of SMN proteins.

The structure of SRSF1 RRM1 bound to the CA dinucleotide showed that the H bond formed between the carbonyl of the Glu87 side chain and the amino group of the cytosine was critical for the specific recognition of the nucleotide. In addition, this glutamate prevented the binding of the protein to a uridine at this position as it would then repulse the carbonyl group present at

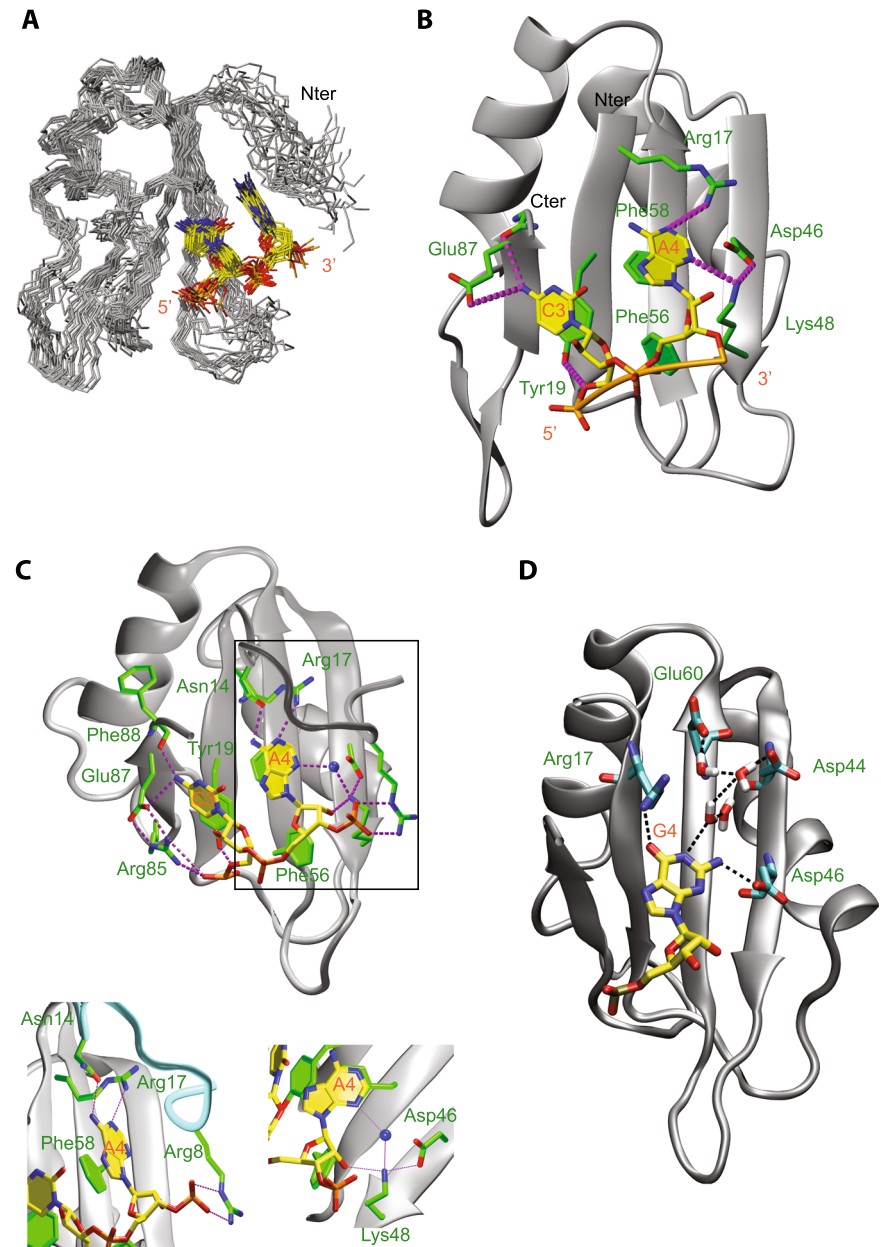

**Fig. 2 Mode of interaction of SRSF1 RRM1 with a CA dinucleotide. A** Overlay of the 20 lowest-energy structures superimposed on the backbone of the structured parts of the protein and heavy atoms of RNA. The protein backbone is shown in gray and heavy atoms are shown in orange (P atoms), yellow (C atoms of RNA), red (O atoms), and blue (N atoms). The RRM (residues 1–89) and the ordered region of RNA (C3 and A4) are shown. **B** The solution structure of the complex is shown in ribbon (protein backbone) and stick (RNA) representation. Protein side chains or backbone involved in RNA interactions are shown as sticks. C atoms of the protein are in green and H bonds in magenta. **C** Snapshot of protein–RNA interactions observed during MD simulations performed with the complex formed by SRSF1 RRM1 and the AACAAA RNA. The blue sphere corresponds to a water molecule. **D** Snapshot of protein–RNA interactions observed during MD simulations performed with the complex formed by SRSF1 RRM1 and the AACGAA RNA.

position 4 of $C_3$. This observation suggested that the replacement of this glutamate by an asparagine could still accommodate the binding of a cytosine and facilitate the recognition of a uridine. Indeed, the asparagine side chain contains a carbonyl and amino group that could interact either with the amino group of a cytosine or with the carbonyl group of a uridine, respectively. We tested this hypothesis in vitro by titrating an SRSF1 E87N RRM1 variant with polyC and polyU sequences. Whereas the wild-type (WT) protein interacted only with polyC (Fig. S1A), the mutated RRM1 protein could also bind to uridines with similar chemical-shift perturbations (Fig. 3A).

MD simulations showed that, in the presence of this mutation, Asn87 side chain could indeed interact with the two bases as initially predicted (Fig. 3B). We then tested whether in the context of the full-length protein this mutation would have the ability to promote SMN2 exon7 inclusion. We co-transfected HEK293 cells with the SMN2 minigene[32] and either the WT or the E87N versions of SRSF1. As expected, the WT protein had an effect similar to the Y19A mutant, which drastically reduces the binding of SRSF1 RRM1 to RNA and therefore prevents SRSF1 to activate SMN2 exon7 splicing. However, in the presence of E87N SRSF1 variant, a clear increase of exon7 inclusion could be

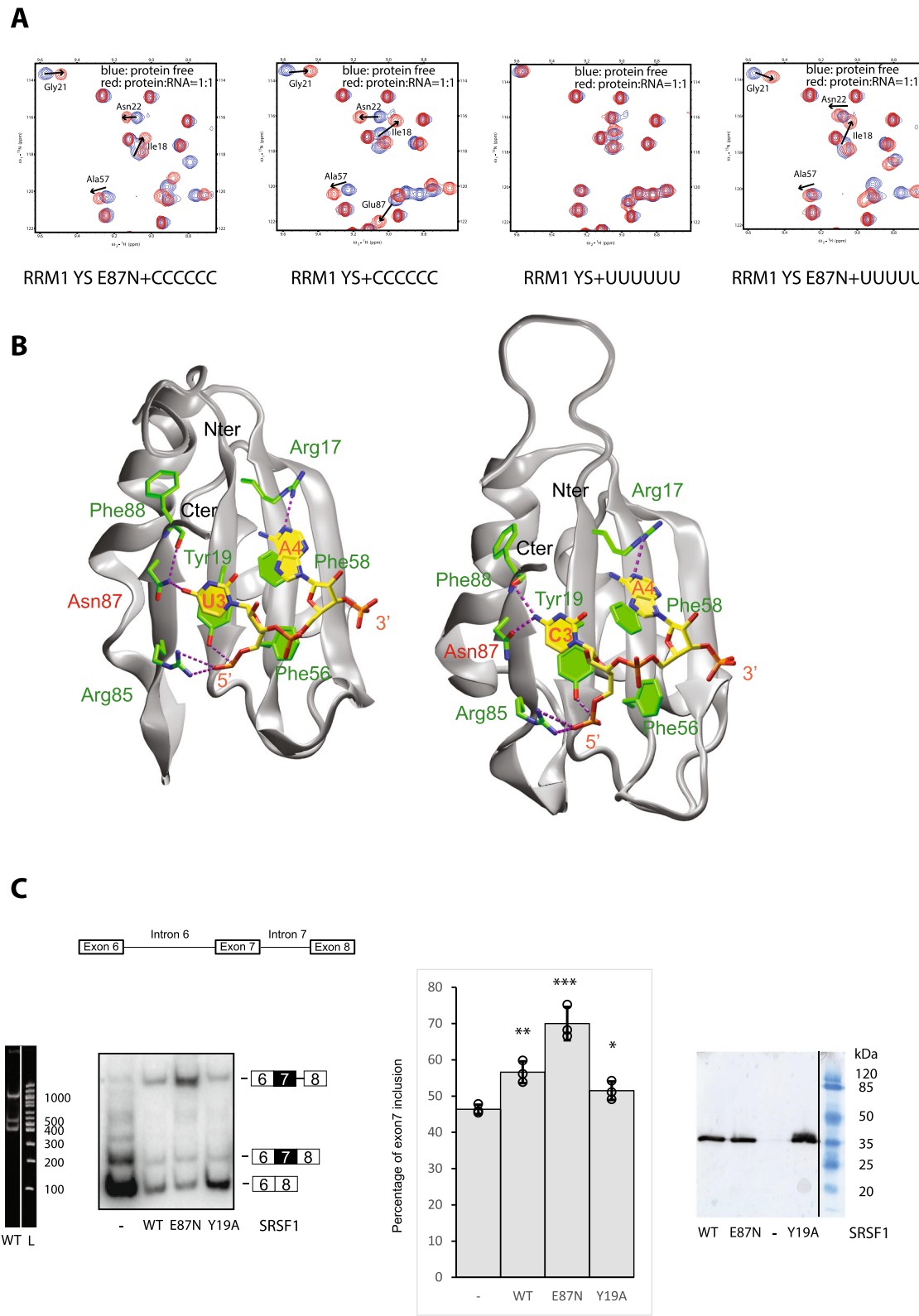

observed although intron7 was still present in splicing products (Fig. 3C).

**SELEX experiment reveals an unexpected bimodal mode of interaction of SRSF1 RRMs with RNA**. We previously found that SRSF1 RRM2 binds to a GGA motif[20] and we now show here that RRM1 recognizes a CN motif. We next wanted to understand

how the two RRMs interact with RNA when linked by their natural interdomain linker and if what we saw with individual domains could be reproduced when the RRMs are in tandem. We then performed a SELEX experiment in the presence of the two RRMs of SRSF1 (SRSF1 RRM1+2) with an RNA containing an invariant GGA motif in the middle flanked by 12 degenerated nucleotides on both sides (Fig. 4A). The idea was to initiate the

**Fig. 3 Engineered SRSF1 E87N binds to uridines and activates SMN2 exon7 inclusion. A** Overlay of $^1$H-$^{15}$N HSQC spectra measured with SRSF1 RRM1 YS and E87N free form (in blue) or in the presence of polyC or polyU molecules at a 1:1 ratio (in red). **B** Snapshot of the protein–RNA interactions observed during MD simulations with the complexes formed by SRSF1 RRM1 E87N and the AAUAAA or AACAAA RNA. The color scheme is the same as in Fig. 2. **C** Schematic representation of the SMN2 minigene. From left to right: 4% acrylamide gel with non-radioactive RT-PCR products showing the levels of SMN2 exon 6-8 isoforms upon overexpression of SRSF1 WT in HEK293 cells and the 100bp ladder of New England Biolabs. The white vertical line indicates the junction between two initially distant lanes present in the same gel. 4% acrylamide gel with radioactive RT-PCR products showing the levels of SMN2 exon 6-8 isoforms upon overexpression of SRSF1 WT, Y19A or E87N in HEK293 cells. Graph showing the percentage of SMN2 exon7 inclusion upon overexpression of WT and mutated versions of SRSF1. Standard deviations and arithmetic means are indicated. Stars indicate $p$ values from two-sided Student's test comparing SRSF1 constructs to the negative control (*$p < 0.05$ (0.036), **$p < 0.01$ (0.0057), and ***$p < 0.001$ (0.001)). Statistics were calculated from the three independent experiments. Data points of each experiment are shown as dot plots. Western blots showing the relative expression levels of the different Flag-tagged versions of SRSF1. The black vertical line indicates the junction between two initially distant lanes present in the same gel.

---

binding of RRM2 to the GGA motif and see whether any sequence enrichment could be observed with the interdomain linker and RRM1. After 6 cycles of selection, a HT-sequencing of all sequences selected with SRSF1 RRM12 was performed. They followed two main patterns containing either the GGA or a GGANGGA motif (Fig. 4B). To determine the positions with the most stringent selection during SELEX, we calculated the relative entropy of position-dependent nucleotide frequency distributions of foreground and background sequences. A clear enrichment in cytosines was observed at positions −4 and +6 of sequences containing a single GGA motif, whereas cytosines were rather selected downstream of the GGANGGA motif (Fig. 4B). This suggests that SRSF1 RRM1 could bind either upstream or downstream of the RRM2-binding site (when only one GGA motif is present), which implies a high flexibility of the inter-RRM linker. The interdomain linker sequence containing nine consecutive glycines would allow such dynamics. To validate this result, we then used two RNA sequences 5'-UCAUUGGAU-3' and 5'-UGGAUUUUUCAU-3' containing the CA motif recognized by the RRM1 at positions −4 and +6 from the two ends of the GGA motif, respectively. In both cases, saturation was reached at a 1:1 ratio and chemical-shift perturbations were observed with similar intensities for both RRM1 and RRM2 amides showing the binding of the two domains to a single RNA molecule (Fig. 4C, D). In good agreement with these NMR data, SRSF1 RRM12 had a similar affinity for the UCAUUGGAU and UGGAUUUUUCAU RNAs ($K_d$ of 58 and 55 nM, respectively), which was similarly decreasing when the CA was mutated to UU in both RNAs ($K_d$ of 164 and 145 nM, respectively) (Fig. S7A). These data also suggest a cooperative mode of interaction of both RRMs with RNA as the $K_d$ values obtained with single RRM1 and RRM2 domains were around 20 and 0.7 µM[20], respectively. Using NMR, we also tested the binding of the two RRMs to the UCAUUGGAUUUUUCAU RNA, which contains two CA motifs at positions −4 and +6 relative to GGA. As shown in the Fig. S7B, saturation was observed at a protein:RNA ratio of 1:1 with the three RNAs tested and the amide chemical shifts observed for SRSF1 bound to the longest RNA were always located between the chemical shifts of the protein bound with the RNA containing a single CA motif either upstream or downstream of GGA. This result indicated that there was no preference for one of the two possible SRSF1-binding conformations on the UCAUUGGAUUUUUCAU RNA. Overall, these results further confirm the sequence specificity of RRM1 for a cytosine but more surprisingly that RRM1 can bind equally well and optimally RNA when the cytosine is present at two fixed positions which are −4 and +6 of the edges of the GGA RRM2 binding site. However, this equal affinity seems to be lost when two consecutive GGA motifs are present in the sequence (Fig. 4B). Indeed, one expect a sliding of RRM2 between these two consecutive GGA motifs[20], which would prevent the binding of RRM1 at the −4 position when RRM2 is bound to the second GGA. As a result, RRM1 is

only binding downstream in such cases. Therefore, the number of GGA motifs would not only increase the affinity of SRSF1 for RNA but also influence the relative position of the two RRMs of SRSF1 on the RNA.

## Discussion

**How NMR structures led to the deciphering of SRSF1 RNA-binding specificity.** From all studies performed in vitro and in vivo with the aim of characterizing the RNA-binding specificity of SRSF1 (Table S2), it was impossible to identify a clear RNA-binding consensus sequence. These data suggested a preference of the protein for binding GA-rich sequences and we previously found that this sequence specificity was coming from the interaction of the pseudo-RRM (RRM2) with a GGA motif[20]. In this study, we revealed that SRSF1 RRM1 also contributes to the specific interaction of the protein with RNA by interacting preferentially with CN motifs. In addition, we showed that the gly-rich interdomain linker allowed the binding of RRM1 to RNA either upstream or downstream of RRM2-binding site at two precise positions (Fig. 4). Indeed, we found that the spacing between the two binding sites may be important for SRSF1 interaction with RNA, as SELEX results indicate the preferred selection of RRM1 binding sites at position −4 or +6 from the edges of the GGA motif (Fig. 4). However, we cannot exclude that RRM1 can also adapt its binding to shorter and perhaps longer distances due to the high flexibility of its inter-RRM linker and of the RNA target, which could form looping structures to enable contacts for distant binding sites. This bimodal mode of interaction was unexpected for SRSF1 and may explain the difficulty of previous investigations to determine a precise RNA-binding motif for this protein. Nevertheless, with such new information, we can now re-examine previously identified binding sites (Table S2). For example, the consensus sequence A**GGA**CAGAG**C** was identified by SELEX with SRSF1 RRM12 and contains a cytosine located 6 nucleotides downstream of the GGA motif[22]. Moreover, an RNA-binding consensus sequence obtained with RNAseq for SRSF1 in the context of breast cancer (U**C**AGA**GGA**)[31] matches well one of the two sequences identified in this study by SELEX (cytosine at position −4 from the GGA motif). Finally, the motif identified in the Krainer's laboratory by functional SELEX as the consensus sequence required for SRSF1 activity in splicing (Ccccc**GG**/cA) also contained the GGA motif bound by RRM2 and was enriched in cytosines with a preference at position −4[33] indicating that our structural data are relevant in cells and for the function of SRSF1 in splicing. In addition, a re-analysis of eCLIP data of SRSF1 (obtained from ENCODE) in two human cell lines (HepG2 and K562) shows a decrease in the frequency of GGA motifs with the rank of the peak, which could well be explained by the predominant function of RRM2 in targeting SRSF1 to its RNA partners in cells (Fig. S8). Indeed, we previously found that SRSF1 RRM2 alone was sufficient to induce the same splicing outcome as the full-length protein in about half of the human splicing

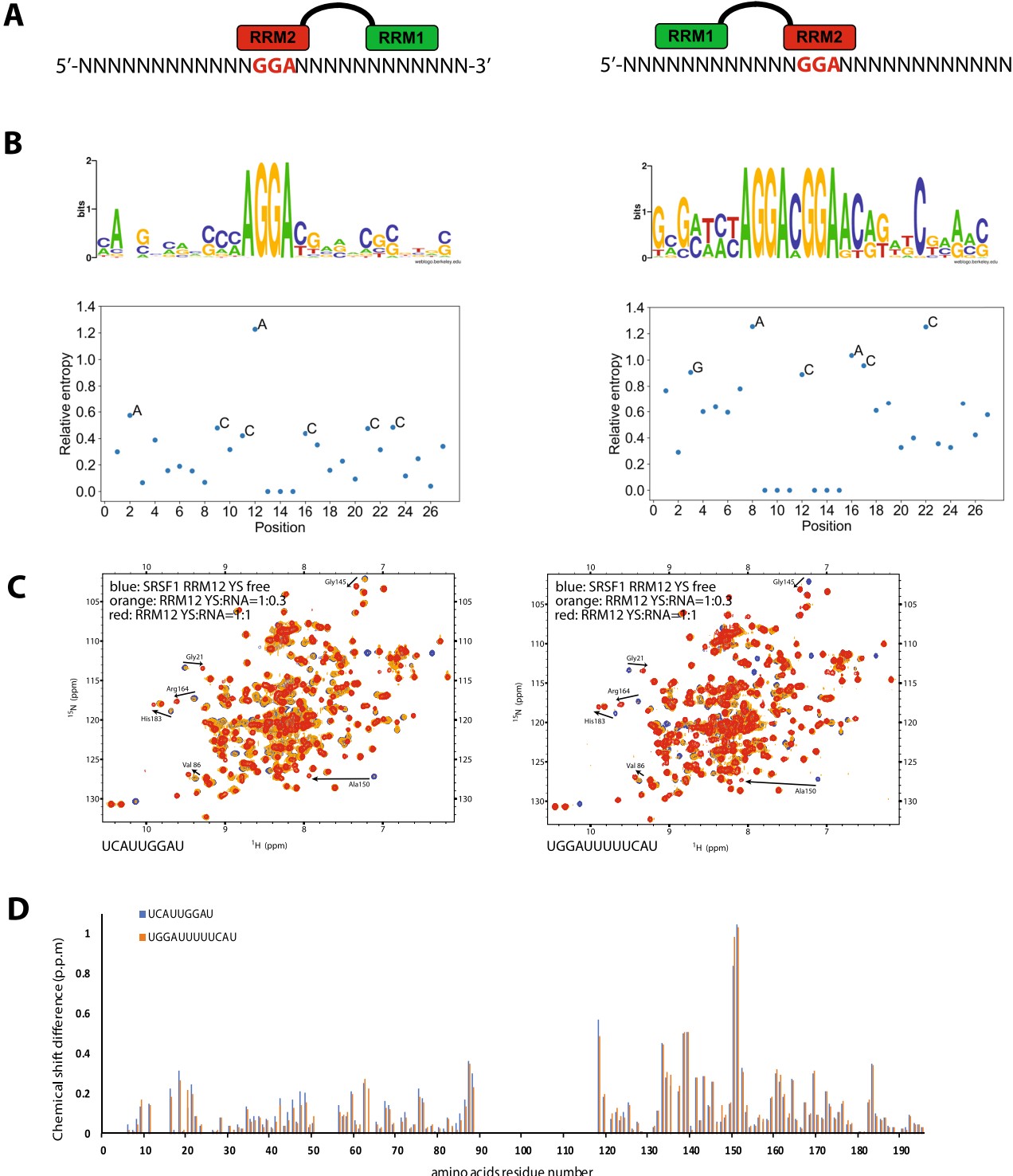

**Fig. 4 SRSF1 RRM1 interacts with RNA on both sides of RRM2-binding site. A** Schematic representation of the RNA molecules used for the SELEX experiment. Invariant GGA nucleotides were inserted in the middle of the degenerated sequence to recruit SRSF1 RRM2 and allow the selection of potential RRM1 binding sites on both sides of the motif. **B** The sequences selected by SELEX with SRSF1 RRM12 followed two main patterns containing either the GGA or a GGANGGA motif. To determine the positions with the most stringent selection during SELEX, we calculated the relative entropy of position-dependent nucleotide frequency distributions of foreground and background sequences (lower panels). **C** Overlay of $^{1}$H-$^{15}$N HSQC spectra measured with SRSF1 RRM1+2 YS free form and in the presence of UCAUUGGAU or UGGAUUUUUCAU RNA at 0.3:1 and 1:1 RNA:protein ratios (in blue, orange, and red, respectively). **D** Mapping of the chemical-shift perturbations observed upon interaction of SRSF1 RRM1+2 YS with the two RNAs tested in **C**. SRSF1 RRM1 binds equally well the cytosine located at −4 or +6 from the GGA motif.

events regulated by SRSF1 that were tested[20]. Additionally, when comparing the 20,000 highest-confidence peaks from each cell line (according to the analysis from the ENCODE consortium) with the 20,000 lowest-confidence peaks, we found that cytosine nucleotides generally flanked the RRM2-binding GGA motifs, but their position was not as clear as in the SELEX data (Figs. 4 and S8). This may be explained by the overrepresentation of GGA motifs in the bound peaks, which makes it difficult to find the register of the sites, but may also point to a more flexible mode of interaction of the two RRMs in the context of the full-length SRSF1 protein than observed in vitro with RRM1+2. Overall, these data suggest that SRSF1 uses a conserved non-conventional and highly flexible mode of interaction with RNA using RRM1 to bind to cytosines either upstream or downstream of RRM2-binding sites. In addition to a direct contact between SRSF1 RRMs and the RRM of the U1-70k component of U1 snRNP[34], SRSF1 was shown to bind to exonic sequences and recruit U2AF35 and U1-70K at the 3'SS and 5'SS, respectively, via an interaction of its RS domain with the RS domain of these two spliceosomal components[35–37]. Interestingly, the mode of interaction that we described here for SRSF1 could well explain this flexibility in recruiting factors either upstream or downstream of SRSF1-binding site.

However, our study also indicates that the specificity of SRSF1 RRM12 for RNA is degenerate. Nevertheless, a previous study suggested that the phosphorylated RS domain could also participate to SRSF1 interaction with RNA by promoting the recruitment of the protein on RNA[15], interacting with the branch point[38], or promoting/stabilizing RNA–RNA duplex formation[39]. In addition, the RS domain could also guide the recruitment of SRSF1 on RNA by interacting with other RS-containing proteins bound at proximity of the RRM-binding sites. A recent study proposed that SRSF1 could induce structural modulations in pre-mRNAs[40]. Our data suggest that SRSF1 could indeed either prevent the formation of stems by maintaining in a single-stranded form the sequence interacting with the RRMs or stabilize potential RNA–RNA duplex or stem formation via the RS domain[39].

We also report here on the unusual differential involvement of the N-terminal extremity of RRM1 depending on the motif bound by the domain. Indeed, this part of the protein is only used to bind a CA motif (Fig. S3). Interestingly, it was previously shown that the deletion of the N-terminal part of SRSF1 permitted RS-domain-independent pre-mRNA splicing[21]. Our results strongly suggest that this effect could originate from the involvement of the domain extremity in binding CA motifs. Moreover, our data can also explain the effect of the phosphorylation of Tyr19 in promoting cell proliferation in pediatric acute lymphoblastic leukemia[41]. Based on the structure of the RRM1 bound to RNA, it is likely that this phosphorylation may prevent the binding of the recognized cytosine by repulsing the negatively charged RNA phosphate of $C_3$ and therefore the binding of SRSF1 RRM1 to RNA (Fig. 3B).

In addition, our structural data help to better understand the mode of RNA recognition of the other SR proteins containing tandem RRMs. We previously found that all pseudo-RRMs were able to bind to GGN motifs (in human, yeast, and fly)[20]. Here we showed that all tested RRM1 in tandem-RRM-containing SR proteins seem to also have a preference for binding cytosines suggesting that there is high conservation in their overall RNA-binding specificity. In good agreement with our predictions, GGN motifs surrounded by cytosine nucleotides were found in the RNA sequences selected with SRSF5 and SRSF6 by SELEX and functional SELEX experiments (Table S2)[24,42,43]. In agreement with a previous study[44], we also showed that the inter-RRM linker could play a key role in RNA recognition by restraining the

relative position of RRM1 and RRM2. The high content in glycines in SRSF1 interdomain linker is not conserved suggesting that, in other SR proteins, RRM1 not necessarily binds on both sides of the RRM2-binding site and optimal distance between the two RRM-binding sites may be different than for SRSF1. This could then explain the difference in binding specificity between the SR proteins. In good agreement with this hypothesis, the sequence of interdomain linkers of SR proteins with tandem RRMs is poorly conserved.

**From structures to the engineering of splicing regulators having different RNA-binding specificities.** We found that SRSF1 RRM1 was responsible for the recognition of the cytosine at position +6 of exon7. Interestingly, a GGU motif is present upstream of this cytosine. Although only two nucleotides instead of the six nucleotides selected in our SELEX experiments are located between the two RRM-binding sites, it may still allow the binding of the RRM2. Indeed, our previous structural investigation on the specificity of SRSF1 pseudo-RRM indicated that a GGU could be accommodated by the domain[20]. The structure of SRSF1 RRM1 bound to RNA allowed us to design a single mutation (E87N) in the β4 strand of the domain, which enables the interaction of the protein with uridines in addition to cytosines. This rationally engineered SRSF1 protein was then able to substantially increase SMN2 exon7 inclusion (Fig. 3C), similarly to the current drug against SMA (SPINRAZA®). Surprisingly, intron7 was still present in the splicing products after E87N variant overexpression in cells (Fig. 3C). This effect is most likely due to the ability of the mutated protein to interact with uridines in addition to cytosines and thereby to bind unexpected additional sequences that are required for the removal of the intron (e.g., binding of E87N SRSF1 to sequences overlapping with binding sites of splicing activators or spliceosomal components). The same effect was recently reported with an ASO targeting SMN2 exon8[45]. Although the stop codon present at the 3'-end of exon7 allows the translation of functional SMN proteins, this intron7 inclusion was shown to introduce negative regulatory elements to the 3' untranslated region inducing a reduction of SMN protein production[45]. The engineering of splicing factors with different RNA-binding specificities is very challenging, but this may represent a promising approach to modulate the splicing outcome in cells. Indeed, a structure-based engineered variant of U2AF65 was also shown to restore the binding of the protein to two mutated splice sites found in human genetic diseases and to increase splicing activity[46]. We showed here that it was also possible to act on splicing regulators without affecting their natural RNA-binding specificity. It suggests that subtle protein mutations guided by structures of protein–RNA complexes may be considered as a therapeutic strategy against the numerous diseases originating from splicing defects.

## Methods
**Preparation of RNA–protein complexes.** We cloned in the pET24 expression vector all RRM1 open reading frames (ORFs) corresponding to amino acids 1–97 of SRSF1, 1–94 of B52, 1–72 of SRSF4 and SRSF6, 1–78 of SRSF5, and 1–88 of SRSF9. These recombinant proteins were fused to an N-terminal GB1 solubility tag followed by 6xHis tag and overexpressed at 37 °C in E. coli BL21 (DE3) codon plus cells in minimal M9 medium containing 1 g/l $^{15}NH_4Cl$ and 4 g/l glucose (for $^{15}N$ labeled protein) or 1 g/l $^{15}NH_4Cl$ and 2 g/l $^{13}C$-glucose (for $^{15}N$- and $^{13}C$-labeled protein). All proteins were purified using two consecutive nickel affinity chromatography (QIAGEN) steps, dialyzed in a 50 mM L-Glu, 50 mM L-Arg, 0.05% β-mercaptoethanol, and 20 mM $Na_2HPO_4$ at pH = 7 buffer and concentrated to 0.8 mM with a 10-kDa molecular mass cutoff Centricon device (Vivascience). GB1-SRSF1 RRM1+2 protein YS (aa 1–196 with the Y37S and Y72S mutations in RRM1) was dialyzed against a buffer that allows the solubility of the two RRM-containing protein at a concentration of 0.2 mM (150 mM KCl, 1.5 mM $MgCl_2$, 0.2 mM EDTA, 50 mM L-Glu, 50 mM L-Arg, 0.05% β-mercaptoethanol, and 20 mM $Na_2HPO_4$ at pH = 7). WT and mutant RNA oligonucleotides were purchased from

Dharmacon, deprotected according to the manufacturer's instructions, lyophilized, and resuspended in the NMR buffer. RNA–protein complexes used to solve structures were formed in NMR buffer at an RNA:protein ratio of 1:1, at a 0.5 mM concentration.

**NMR measurement**. All NMR measurements were performed in the NMR buffer at 313 °K using Bruker AVIII-500 MHz, 600 MHz, 700 MHz, and Avance-900 spectrometers equipped with a cryoprobe. Data were processed using Topspin 3.6.2 (Bruker) and analyzed with Sparky 3.133 (http://www.cgl.ucsf.edu/home/sparky/).

Protein sequence-specific backbone and side chain assignments were achieved using two-dimensional (2D) $^1$H-$^{15}$N heteronuclear single quantum coherence (HSQC), 2D $^1$H-$^{13}$C HSQC, three-dimensional (3D) HNCA, 3D HNCO, 3D CBCACONH, 3D HcCH total correlated spectroscopy (TOCSY), 3D NOESY $^1$H-$^{15}$N HSQC, and 3D NOESY $^1$H-$^{13}$C HSQC aliphatic (for a review, see ref.[47]). Aromatic proton assignments were performed using 2D $^1$H-$^1$H TOCSY and 3D NOESY $^1$H-$^{13}$C HSQC aromatic.

Resonance assignments of RNA in complex with SRSF1 RRM1 were performed using 2D $^1$H-$^1$H TOCSY, 2D $^1$H-$^1$H NOESY, 2D $^{13}$C 1F-filtered 2F-filtered NOESY[48], and natural abundance 2D ($^{13}$C-$^1$H) HSQC in 100% D$_2$O.

Intermolecular NOEs were obtained using 2D $^1$H-$^1$H NOESY, 2D 1F-edited 2F-filtered NOESY, and 3D $^{13}$C-resolved 1F-edited 3F-filtered HSQC-NOESY[49] using unlabeled RNA and $^{15}$N and $^{15}$N-$^{13}$C-labeled proteins, respectively. All NOESY spectra were recorded with a mixing time of 150 ms, the 3D TOCSY spectrum with a mixing time of 23 ms, and the 2D TOCSY with a mixing time of 50 ms.

**Structure calculation and refinement**. AtnosCandid 2.1 software[50,51] was used to generate preliminary structures and a list of automatically assigned NOE distance constraints for SRSF1 RRM1 in complex with RNA. Peak picking and NOE assignments were performed using 3D NOESY ($^{15}$N- and $^{13}$C-edited) spectra. Additionally, intraprotein hydrogen bond constraints were added based on hydrogen–deuterium exchange experiments on the amide protons. For these hydrogen bonds, the oxygen acceptors were identified based on preliminary structures calculated without hydrogen bond constraints.

Seven iterations were performed and 100 independent structures were calculated at each iteration step. Structures of the protein–RNA complexes were calculated with CYANA 3.98.13[51] by adding the manually assigned intramolecular RNA and RNA–protein intermolecular distance restraints. For each CYANA run, 50 independent structures were calculated. These 50 structures were refined with the SANDER module of AMBER 12[52] by simulated annealing run in implicit water using the ff99 force field.

The best structures based on energy and NOE violations were analyzed with PROCHECK 3.5.4[53]. The Ramachandran plot of the SRSF1 RRM1 in complex with RNA indicates that 77.8% of the residues are in the most favored regions, 21% in the additional allowed regions, 0.7% in the generously allowed regions, and 0.5% in the disallowed regions. All the figures showing structures were generated with MOLMOL 2K.2[54].

**SELEX**. A DNA matrix containing a central GGA motif with 12 degenerated nucleotides on each side was produced by DNA pol I elongation (Biolabs) for 1 h at 37 °C using two partially complementary oligonucleotides (5'-CGCGAATTCtaatac gactcactataGCGCCGACCAACGACATT-3' and 5'-GCGCTCGAGATGGGCACT ATTTATATCAACNNNNNNNNNNNNNTCCNNNNNNNNNNNNNAATGTCGT TGGTCGGCGC-3') containing a T7 RNA polymerase promoter (lower case). After sodium acetate precipitation, 10 µg of DNA was used as a template for in vitro transcriptions with 20 mM MgCl$_2$, 6 mM rNTPs, 4 mM GMP, 5× dimethyl sulfoxide in TB buffer (40 mM Tris HCl pH 8.1, 1 mM spermidine, 0.01% Triton X-100, 5 mM dithiothreitol) for 3 h at 37 °C. RNA was purified on high-performance liquid chromatography, butanol extracted, and incubated with 100 µl of Ni-NTA agarose beads slurry (QIAGEN) in the absence of protein to eliminate RNA molecules having some affinity for the beads. Twenty nmoles of eluted RNA was incubated for 30 min at 4 °C with His-tagged GB1-SRSF1 RRM1+2 protein at a 1:20 RNA:protein ratio in 300 µl of buffer D (20 mM HEPES-KOH pH 7.9, 150 mM KCl, 1.5 mM MgCl$_2$, 0.2 mM EDTA, 10% glycerol) and added to 100 µl of Ni-NTA agarose bead slurry (QIAGEN) for 2 h at 4 °C. Beads were then washed 5 times with buffer D containing 0.05% of NP40 and incubated with 100 µg of proteinase K for 15 min at 37 °C. After phenol–chloroform extraction, selected RNAs were precipitated with 0.3 M sodium acetate in the presence of 1 µl of glycogen and reverse-transcribed using a reverse primer (5'-GCGCTCGAGATG GGCACTATTTATATCAAC-3') and the reverse transcriptase Superscript III (Invitrogen) for 50 min at 50 °C.

Then 30 cycles of PCR amplification were performed in the presence of two primers (5'-CGCGAATTCTAATACGACTCACTATAGCGCCGACCAACGACA TT-3' and 5'-GCGCTCGAGATGGGCACTATTTATATCAAC-3') (50 pmol each), 4 mM MgSO$_4$, 0.4 mM dNTPs, and 1.25 units of Taq DNA polymerase (Biolabs). The amplified DNA fragments were then used as a new matrix for in vitro transcription and additional cycle of selection. After six cycles of amplification–selection experiments, DNA fragments were tagged and submitted to HT-sequencing.

**Isothermal titration calorimetry**. ITC experiments were performed on a VP-ITC instrument (Microcal), calibrated according to the manufacturer's instructions. Protein and RNA samples were dialyzed against the NMR buffer. Concentrations of proteins and RNAs were determined using optical density absorbance at 280 and 260 nm, respectively. Ten µM of each RNA were titrated with 300 µM of GB1-SRSF1 RRM1 YS protein and 10 µM of GB1-SRSF1 RRM12 YS protein was titrated with 100 µM of each RNA by 40 injections of 6 µl every 5 min at 40 °C. Raw data were integrated, normalized for the molar concentration, and analyzed using the Origin 7.0383 software according to a 1:1 RNA:protein ratio binding model.

**Bioinformatics analysis**

*Analysis of SELEX data*. We sequenced a total of 14,500,301 unique sequences and 28,192,850 unique sequences obtained after 6 rounds of selection. In the input set, there were 7044 sequences that occurred in >1 copy, whereas in the SELEX-selected set, 1,605,630 sequences had >1 copy. To uncover the positions which experienced the most selection, the nucleotide composition of the 212 sequences with >100 copies in the SELEX-selected set (referred to as "foreground" sequences) relative to that of sequences with at most 10 copies in the input data set (referred to as "background") were analyzed. Upon inspection, we found that the foreground sequences followed two main patterns: 94 of the 212 unique sequences had GGANGGA at positions 29–35, while 98 sequences had only a GGA at positions 33–35 (occurrences of the GGA motif at other positions in these sequences were not excluded but did not follow a position-specific pattern). We used the weblogo software[55] to generate position-specific weight matrices representing these sequences. Furthermore, to determine the positions with the most stringent selection during SELEX, we calculated the relative entropy of position-dependent nucleotide frequency distributions of foreground and background sequences selected as described above. That is, if $F_i(\alpha)$ is the nucleotide frequency distribution at position $i$ in the foreground and $B_i(\alpha)$ the corresponding distribution in background sequences, with $\alpha \in$ (A,C,G,T), the relative entropy at position $i$ was calculated as $\Sigma_\alpha F(\alpha) * \log(F(\alpha)/B(\alpha))$.

The binding peaks from the SRSF1 eCLIP data for the HepG2 and the K562 cell lines[56] were obtained from ENCODE[57], in the form of BED files (ENCFF214HOA, ENCFF508DKJ, ENCFF373NZF, ENCFF205JTY). There are two replicates per cell type. For each of these replicates, the peaks are sorted according to their $p$ value. The center of each peak is used and extended by 25 nucleotides on each side, and then peaks that overlap by one or more nucleotides are merged. The $p$ value of the merged peak is the most significant of the $p$ values of the merged peaks. Non-overlapping sets of 1000 peaks, starting from the most to the least significant, are then used to calculate the average frequency of GGA motifs per position per peak, first calculating the number of GGAs detected in each peak divided by the peak length and then averaging over 1000 peaks at a time (Fig. S8A).

For the positional weight matrices (PWMs), the best 10,000 peaks according to their $p$ value are kept for each of the replicates, yielding 20,000 peaks per cell type. For each peak where a GGA is found, the 12 nucleotides to the left and to the right of the GGA motif are obtained. If a peak contained more than one GGA, each of these GGAs was used as a separate center. The sequences extracted from the left and right of each GGA are used for producing a PWM (Fig. S8B, C). A similar procedure is used to extract the 10,000 lowest significance peaks from each replicate, which were then used as "background" for calculating the relative entropy of position-dependent nucleotide frequencies around GGA motifs.

**Molecular dynamics**. We have used the first frame of the structural ensemble of the SRSF1 RRM1 protein/RNA complex (this work) to start all MD simulations. The structures containing the E87N mutation and the C$_3$ to U$_3$ and A$_4$ to G$_4$ nucleotide replacements, respectively, were prepared by molecular modeling. AMBER16 was used to prepare the initial files and to execute the simulations. We have used bsc0χ$_{OL3}$[58] and ff12SB[59] force fields to describe the RNA and protein, respectively. The systems were simulated in octahedral box of explicit SPC/E[60] water molecules with minimal distance of 12 Å between solute and the box border and KCl ions[61] were used to neutralize the system, achieving an excess salt concentration of 0.15 M. The NMR restraints were utilized in initial part of the simulations to improve stability of the subsequent unbiased simulations[62]. Details of the simulation equilibration and production protocol were discussed previously[63]. We have performed four simulations of the WT RRM1 bound to AACAAA RNA, which corresponds to the solution structure presented in this work. We have further performed four simulations of the WT RRM1/AACGAA and two and three simulations of the E87N RRM1/AACAAA and E87N RRM1/AAUAAA systems, respectively. The length of the simulations was 1 µs. One WT RRM1/CA and one WT RRM1/CG simulation were extended to 10-µs timescale to more thoroughly examine the N-terminus/RNA interactions. VMD 1.9.2 was used to visualize MD trajectories.

**Cell culture and plasmids**. Human embryonic kidney HEK293 cells were maintained in Dulbecco's modified Eagle's medium (GibcoBRL) supplemented with 10% fetal calf serum (GibcoBRL). The mammalian vectors expressing FLAG-tagged SRSF1 WT, Y19A, and E87N were obtained by cloning the SRSF1 ORF, amplified by PCR, in frame into the pCAG vector (Addgene). SRSF1 Y19A and E87N mutants were created by site-directed mutagenesis using specific primers.

**In vivo splicing assays**. HEK293 cells were plated on six-well plates. For each transfection, 1 μg of pCI-SMN2[64] (WT or mutants) plasmid was cotransfected with 3 μg of pcFLAG-SRSF1 (WT or mutants) using the calcium phosphate method. Equal amounts of DNA were transfected by addition of empty vector when necessary. Forty-eight hours after transfection, total RNA was isolated. One microgram of total RNA was reversed transcribed using oligo(dT) and Superscript II (Invitrogen) or M-MuLV Reverse Transcriptase RNAseH⁻ (Finnzyme). One-tenth of the cDNA was amplified by PCR using a vector-specific forward primer (pCI-forw: 5'-GGTGTCCACTCCCAGTTCAA) and the SMN-specific reverse primer SMNex8-rev (SMNex8-rev: 5'-GCCTCACCACCGTGCTGG). The SMNex8-rev primer was $^{32}$P 5'-end labeled and PCR reactions were terminated during the linear phase (24 cycles; 94 °C, 30 s; 55 °C, 30 s; 72 °C, 60 s). After electrophoresis on a 4% polyacrylamide gel, the ratio of exon inclusion (SMN2-FL) to exon skipping (SMN2Δ7) was determined by using AlphaView (proteinsimple, San Jose, CA). Means and standard error of the mean were calculated from the three independent experiments.

**Western blotting**. Protein samples were separated on a 12% sodium dodecyl sulfate–polyacrylamide gel electrophoresis and then electroblotted onto nitro-cellulose membranes (Whatman Optitran BA-S 83), blocked with 5% non-fat dry milk in Tris-buffered saline + 0.1% Tween (TBS-T). Probing of the blots was done with monoclonal ANTI-FLAG M2 Antibody (mouse) from Sigma (F1804) 1 μl diluted 1000× in 5 ml of TBS-T buffer and 5% non-fat dry milk, followed by incubation with 1 μl horseradish peroxidase-conjugated anti-mouse (Sigma) for 2 h in 10 ml (1/10,000) of TBS-T and 5% non-fat dry milk. Protein signals were detected with chemiluminescence imaging (Amersham Imager 600RGB).

**Reporting summary**. Further information on research design is available in the Nature Research Reporting Summary linked to this article.

## Data availability

We have deposited the coordinates of the SRSF1 RRM1 AACAAA structures in the protein Data Bank (PDB) under the PDB ID 6HPJ. Other data and materials are available from the authors upon reasonable request. Source data are provided with this paper.

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

## Acknowledgements

We thank Dr. Alvar Gossert, Dr. Simon Rudisser, and Dr. Fred Damberger for the maintenance of the biomolecular NMR spectroscopy platform and Prof. Constance Ciaudo and Dr. Richard Patryk Ngondo for the access to the Amersham Imager 600RGB. We also thank the SNF-NCCR RNA and Disease (grant number: 51NF40-182880), the sinergia (grant numbers: CRSII5_170976 and CRII3_141942), SMA Europe, cure-SMA, and the "Fondation Suisse de Recherche sur les maladies musculaires" for financial support to F.H.-T.A. and A.C. M. Krepl and J.S. acknowledge support by Czech Science Foundation project 20-16554S.

## Author contributions

A.C. and F.A. were involved in the determination of the structure; C.K.X.N. and A.C. performed NMR titrations; A.C. performed the ITC measurements; A.M. and A.C. performed in vivo splicing assays; A.C. performed the SELEX experiment; H.J., M. Katsantoni, M.O., N.M., and M.Z. performed all the bioinformatics analysis, M. Krepl and J.S. were in charge of the MD study. All the authors contributed to the redaction of the manuscript.

## Competing interests

The authors declare no competing interests.
