## [Peer Review File · Nature Communications]

REVIEWER COMMENTS

Reviewer #1 (Remarks to the Author):

The manuscript “Structure of SRSF1 RRM1 bound to RNA reveals an unexpected pendular/bimodal mode of interaction and explains its involvement in SMN1 exon 7 splicing” by Cléry et al., reports an extremely insightful study about the RNA recognition of SRSF1 RRM1 domain. The authors could combine this new data with data from their previous work about pseudo-RRM2 and could provide a convincing model of how the tandem domains bind to certain splice sites and how this can be modulated. Of special interest is the mutation (E87N), which modulates base specificity and thus changes SMN2 splicing. This could be indeed a potential alternative to available drugs against SMA. The manuscript is well written (apart from a few typos) and concise and the discussion is imaginative but not speculative. After the comments listed below are satisfyingly answered, this manuscript clearly merits publication in Nature Communications.

A major concern I have is with the surprising lack of quantifying affinities. Although the author’s qualitative assessment of binding strength and therefore base preference is convincing in most cases, I think it would be quite straightforward to obtain dissociation constants from their NMR data, considering the quality of NMR data and that all RRM1-RNA interactions seem to be in the fast exchange regime (with regards to the single RRM titrations). Also ITC data would be quite easy to acquire it seems.

This is especially important for their comparison of RRM1 wt and E87N mutant and how this mutation modulates binding to polyC and polyU (please add S1A (C and U) to Figure 3A also. Easier for the reader to compare. Also, in each hsqc spectrum and zoom view of a titration, I miss the stoichiometries. They are sometimes given in the figure legends, but it is easier to see this directly in the figure. Determination of a dissociation constant would be also important when comparing AACAAA and AACGAA. Following this, it would be very illustrative to model a G instead of the A in their structure to assess whether other contacts are formed, which could explain the same or even higher affinities.

The authors claim that they can observe the C preference for all related proteins they tested. However, from the figures they show, I think this is an overstatement when looking at B52 and SRSF4. This part should be slightly rewritten.

Please provide the CSP histogram plots for all titrations below the spectrum. The figure quality also should be improved (stoichiometries, labels, residue labels, residue numbering according to native sequence...if it is not at the moment).

The author’s derive from their SELEX data that the linker is very flexible to allow binding of the CA motif on both sides of the GGA. This is convincing, but could be easily further confirmed with 15N relaxation data of unbound tandem RRM1-RRM2. From this it could be also assessed whether both domains tumble totally independent from each other in the free state.

It would be interesting to test an oligo where a CA is on both sides (-4 and +6), to see if there is a preference, or if the domain swaps back and forth. The overall affinity should be even higher?

Minor concerns:

On line 141, I would complete the sentence with 33 intermolecular NOE-derived distance restraints between protein and RNA, to make it clearer for a wider readership.

Reviewer #2 (Remarks to the Author):

This is an important and timely study that focuses on understanding the structure and function of human SRSF1, an RNA binding protein important for RNA metabolism and implicated in oncogenesis. The protein contains two RNA recognition motifs (RRM1 and RRM2), of which RRM2 had previously been characterized by this lab. The present studies reveal that RRM1 bind promiscuously to RNA "CN" sequences (N = any nucleotide). Based on the NMR structure of an RRM1:RNA complex, the authors engineered a mutant protein that enhances affinity for uridine-containing RNAs and thereby activates SMN2 exon inclusion, suggesting a potential avenue for using SRSF1 mutants as a therapeutic strategy for treatment of spinal muscular atrophy.

The experiments used to reach the conclusions are all state-of-the-art and the data presented are of high quality, as expected from the Allain group. The paper is well-written; I was unable to find a single typo. My only suggestion is that SMN2 be defined (or just deleted; I don't think it is necessary) in the abstract.

Reviewer #3 (Remarks to the Author):

The authors provide a strong experimental validation combining NMR and MD simulation data to previously asserted binding of a rather dynamic RRM1 of SRSF1 to C nucleotides containing ssRNA through a canonical β -sheet surface (and indicates the minimal motif to be CN). They also have extended their observations to RRM1 of four other two-RRM-containing SR proteins namely SRSF4, SRSF5, SRSF6, and SRSF9 of humans and SRp55 of Drosophila. They further highlight the possibility of two distinct interaction specificity of RRM1 and RRM2 (joined with a flexible linker) to engage in a bimodal interaction and thus providing a complex variability. They also use this discovery to identify the mechanistic reason for the well-characterized phenomenon that SRSF1 cannot interact with the ESE1 of SMN2 exon 7. This absence of interaction, caused by a C-to-U transition in the exon 7 of SMN2 in comparison to that of SMN1, causes skipping of exon 7 of SMN2. Before moving onto the solution structure of RRM1, they identified the residues on RRM1 that cause aggregation of SRSF1-RRM1. The solution structure aids in the identification of the amino acid residues of RRM1 involved in the

interaction with RNA. Based on this information, the authors introduced mutations in RRM1 of full-length SRSF1 and showed that this mutant is capable of restoring inclusion of exon 2 of SMN2 in vivo. They also proved their rationale for this mutation in vitro by showing that this mutation enhances the interaction of the protein to 'U'-rich sequences in addition to 'C'-rich sequences. Then they carried out an ingenious SELEX experiment to establish the bimodal interaction of SRSF1 involving the affinity for CN of the RRM1 and for GGN of the RRM2.

This work is timely and has been long overdue as the authors have pointed out. Most importantly, this work is to be credited for its contribution to basic understanding of the mechanism of action of one of the most studied and fundamental group of splicing factors – the SR proteins. The extension of the observation to five SR proteins is notable. However, I believe, the presentation of the manuscript should be more focused and should attempt to discuss some of the other aspects of SRSF1 that have recently been unraveled.

Major points:

Can it be distinguished (while assessing the importance of the nucleotide at the second position by titrating SRSF1 RRM1 with NNCGNN, NNCCNN, NNCTNN or NNCANN) whether the effect is not coming from flanking C nucleotides? Perhaps should have been compared with N=A, G and T.

Is there any data suggesting both RRMs are binding simultaneously to GGA containing SELEX RNA to justify their argument of a flexible linker between 2 RRM motifs? Binding affinity data for sequences 5'-UCAUUGGAU-3' and 5'-UGGAUUUUUCAU-3' containing the CA motif are missing.

In general, binding affinity analysis of RRM1/2 with a select set of RNA with both RRM binding sites will make the manuscript lot stronger.

Can it be separated whether the RNA is binding to both at the same time or in succession by NMR experiment? Overlay of 1H-15N HSQC spectra measured with SRSF1 RRM1+2 YS free form and in the presence of UCAUUGGAU or UGGAUUUUUCAU RNA..

The re-analysis of mouse and human (K562 and HepG2) CLIP data needs to be analyzed to check presence and distribution of C in more detail to strengthen the utility of this manuscript.

The authors have identified two aspects of SRSF1 biochemistry – its interaction with RNA and its self-aggregation. However, they have not fully utilized this information to put forward models to explain known functions and behaviors of SRSF1. A recent publication (Nucleic Acids Res 2020 Jun 19;48(11):6294-6309) suggests that SRSF1 binds cooperatively to the pre-mRNA depending on availability of single-stranded sequence immediately upstream of the 5'SS, which then structurally remodels the pre-mRNA. This cooperativity could be explained by the interaction the authors have identified. Optionally the authors might want to use their Y37/Y72 mutants to examine if the cooperativity of the full-length proteins is caused by these residues. Additionally, the authors should

invoke a discussion how self-aggregation and bimodal interaction help SRSF1 in remodeling the pre-mRNA.

The authors must include the RS domain of SR proteins in their discussion. RS domains are phosphorylated for activating SR proteins. It is also known to bind the branchsite. Although RS domain is not essential for splicing of several model pre-mRNA substrates tested in vitro, its presence in vivo warrants its inclusion while discussing sequence specificity of SRSF1 binding, particularly when the binding consensus is so degenerate.

Minor Points:

The author observed the limitation of their mutation in rectifying the splicing of SMN2 gene (inclusion of exon 7 is accompanied by retention of intron 7 – Figure 3C). Yet, they advanced the idea of this mutation as a potential cure for SMA. This SRSF1 mutant being able to rescue skipping of exon 7 is a sufficient advancement of splicing biochemistry. This does not have to be (weakly) established as a cure for SMA. Therefore, the claim that this mutation could be a potential cure should be softened or possibly removed altogether from the manuscript.

Too many glamor words- unexpected, bimodal/pendular, conserved non-conventional etc. The work is informative but bimodal interaction is not exactly novel. Neither is the recognition of C by RRM1.

The fact that RRM1 binds to a distinct motif was known/envisioned but has not been investigated, it does not necessarily explain the difficulty to derive a consensus binding sequence for this protein so far.

-Figure 1A schematic needs better representation

A lack of clear definition of the interaction since restraints not sufficient to describe the RNA binding interface precisely and MD simulation was used.

4E model is far-fetched.

REVIEWER COMMENTS

Reviewer #1 (Remarks to the Author):

The manuscript “Structure of SRSF1 RRM1 bound to RNA reveals an unexpected pendular/bimodal mode of interaction and explains its involvement in SMN1 exon 7 splicing” by Cléry et al., reports an extremely insightful study about the RNA recognition of SRSF1 RRM1 domain. The authors could combine this new data with data from their previous work about pseudo-RRM2 and could provide a convincing model of how the tandem domains bind to certain splice sites and how this can be modulated. Of special interest is the mutation (E87N), which modulates base specificity and thus changes SMN2 splicing. This could be indeed a potential alternative to available drugs against SMA. The manuscript is well written (apart from a few typos) and concise and the discussion is imaginative but not speculative. After the comments listed below are satisfyingly answered, this manuscript clearly merits publication in Nature Communications.

Thank you for your positive feedback on our work and your comments that we took into account for the revision of this manuscript. To answer to your comments, we performed several additional experiments: new NMR titrations, ITC measurements with 8 different RNAs and new MD simulations. All modifications performed in the text are in red. Please find our answer below to each of your comment.

A major concern I have is with the surprising lack of quantifying affinities. Although the author’s qualitative assessment of binding strength and therefore base preference is convincing in most cases, I think it would be quite straightforward to obtain dissociation constants from their NMR data, considering the quality of NMR data and that all RRM1-RNA interactions seem to be in the fast exchange regime (with regards to the single RRM titrations). Also ITC data would be quite easy to acquire it seems.

This is especially important for their comparison of RRM1 wt and E87N mutant and how this mutation modulates binding to polyC and polyU (please add S1A (C and U) to Figure 3A also. Easier for the reader to compare. Please provide the CSP histogram plots for all titrations below the spectrum. The figure quality also should be improved (stoichiometries, labels, residue labels, residue numbering according to native sequence...if it is not at the moment). Also, in each hsqc spectrum and zoom view of a titration, I miss the stoichiometries. They are sometimes given in the figure legends, but it is easier to see this directly in the figure. Determination of a dissociation constant would be also important when comparing AACAAA and AACGAA.

We modified the figures according to the comments of the reviewer. In addition, we performed ITC titrations of the SRSF1 RRM1 with AACAAA and AACGAA RNAs and obtained similar Kd values (21 and 25 μ M, respectively) (Fig. S4C). We also tested the binding of the protein to CCCCCC and UUUUUU RNAs and, as expected from our NMR analysis, observed a binding only with the polyC RNA. The affinity is slightly higher than with the AACAAA and AACGAA RNAs (Kd of 11 μ M instead of 21-25 μ M), which is most likely due to an avidity effect (Fig. S4C).

We also tested with ITC the binding of the SRSF1 E87N RRM1 mutant to the polyC and polyU RNA but the protein was not behaving properly and even after contacting an expert from Malvern (the company providing the ITC instruments), the data that we obtained could not be fitted. However, based on the intensity of the chemical shift perturbations that are now compared in Figure 3A, we can say that the affinity of SRSF1 RRM1 E87N for polyC and polyU sequences is in the same range, although slightly lower than for SRSF1 RRM1 YS with

polyC. The exact affinity obtained *in vitro* for this domain would anyway be difficult to interpret as the affinity increases in the context of the 2 RRMs (see below) and most likely even more in the presence of the RS domain. The important message that we try to convey in the text is the difference in specificity between RRM1 WT and E87N, which is clear from the chemical shift perturbations observed with NMR. It shows that it is possible to modulate the mode of RNA recognition of splicing regulators like SRSF1 with an equivalent efficiency as shown by the overlay of the spectra obtained with polyC and polyU (Fig. 3A).

The text was updated accordingly to these new results.

Following this, it would be very illustrative to model a G instead of the A in their structure to assess whether other contacts are formed, which could explain the same or even higher affinities.

As suggested by the referee, we have modelled the protein-RNA interaction with a CG instead of CA-containing RNA using thirteen microseconds of additional MD simulations to more thoroughly examine the N-terminus/RNA interactions. The simulations showed that instead of interactions with the N-terminus (seen for A₄), the G₄ formed interactions with the RRM domain residues. These interactions were stable and did not allow any contacts between the N-terminus and G₄. This result provides a rationale for the lack of chemical shift perturbations in the N-terminal region upon binding of CG-containing RNA. It also explains why the affinity is not reduced since the loss of interactions with the N-terminus is compensated by interactions with the main body of the RRM. In addition to the text detailing our new MD simulations, we have updated the Material and Methods section. A new figure detailing the SRSF1 RRM1 interactions with the CG-containing RNA was added as a panel D in Figure 2.

The authors claim that they can observe the C preference for all related proteins they tested. However, from the figures they show, I think this is an overstatement when looking at B52 and SRSF4. This part should be slightly rewritten.

We changed the sentence:

“Interestingly, we obtained the same trend with the RRM1 of the human SRSF4, SRSF5, SRSF6, SRSF9 and *Drosophila* B52 (Fig. S2) suggesting that this preferential binding to cytosines is conserved in the RRM1 of all SR proteins containing two RRMs.”

By

“Interestingly, we obtained the same trend with the RRM1 of the human SRSF4, SRSF5, SRSF6, SRSF9 and *Drosophila* B52 (Fig. S2) suggesting that this preferential binding to cytosines **may be conserved in other RRM1 of SR proteins** containing two RRMs.”

The author's derive from their SELEX data that the linker is very flexible to allow binding of the CA motif on both sides of the GGA. This is convincing, but could be easily further confirmed with ¹⁵N relaxation data of unbound tandem RRM1-RRM2. From this it could be also assessed whether both domains tumble totally independent from each other in the free state.

In this manuscript, we do not focus on the free form of SRSF1 RRM12, but on the ability of the two RRMs to bind RNA with RRM1 upstream or downstream of RRM2, which is clear from the NMR titrations shown in Fig 4C (same chemical shift perturbations for RRM1 and RRM2 amides with both RNAs and saturation at a 1:1 ratio in both cases). We are currently trying to obtain structures of the free and bound forms of the two RRMs bound to RNA and will then reveal how the transition from one to the other form is possible. We completely agree with the referee that ¹⁵N relaxation data will then be required to fully understand this

process. However, presenting them without any additional structural support will be misleading and out of the scope of the present work.

It would be interesting to test an oligo where a CA is on both sides (-4 and +6), to see if there is a preference, or if the domain swaps back and forth. The overall affinity should be even higher?

Yes, we agree that this is an interesting question. We prepared a figure with an NMR overlay of SRSF1 RRM12 bound to UCAUUGGA, UGGAUUUUUCAU and UCAUUGGAUUUUUCAU RNAs (Fig S7B). Saturation was obtained at a 1:1 ratio for all RNAs and the chemical shifts observed at this ratio with the UCAUUGGAUUUUUCAU RNA were located in between those observed with UCAUUGGA and UGGAUUUUUCAU RNAs. This result suggests the presence of an exchange between the two populations or at least no preference for one of the two binding modes, which is in agreement with affinities that we measured in addition to the request of the reviewer:

Indeed, using ITC, we also investigated the interaction of SRSF1 RRM12 with the UCAUUGGAU and UGGAUUUUUCAU RNAs and observed a similar affinity (58 and 55 nM, respectively) (Fig. S7A). Finally, we tested the binding of the protein to RNA mutants in which the CA motif was mutated to UU and obtained a decrease in affinity for both RNAs indicating clearly a contribution of the RRM1 in the binding and a cooperative interaction when both RRMs are linked by their natural linker (Kd values of 164 and 143nM with UUUUUGGAU and UGGAUUUUUUU RNAs, respectively).

Minor concerns:

On line 141, I would complete the sentence with 33 intermolecular NOE-derived distance restraints between protein and RNA, to make it clearer for a wider readership.

The sentence was modified accordingly to the reviewer comment.

Reviewer #2 (Remarks to the Author):

This is an important and timely study that focuses on understanding the structure and function of human SRSF1, an RNA binding protein important for RNA metabolism and implicated in oncogenesis. The protein contains two RNA recognition motifs (RRM1 and RRM2), of which RRM2 had previously been characterized by this lab. The present studies reveal that RRM1 bind promiscuously to RNA "CN" sequences (N = any nucleotide). Based on the NMR structure of an RRM1:RNA complex, the authors engineered a mutant protein that enhances affinity for uridine-containing RNAs and thereby activates SMN2 exon inclusion, suggesting a potential avenue for using SRSF1 mutants as a therapeutic strategy for treatment of spinal muscular atrophy.

The experiments used to reach the conclusions are all state-of-the-art and the data presented are of high quality, as expected from the Allain group. The paper is well-written; I was unable to find a single typo. My only suggestion is that SMN2 be defined (or just deleted; I don't think it is necessary) in the abstract.

We are pleased that the referee appreciated our work. Thank you so much!
We replaced SMN2 by SMN to simplify the abstract.

Reviewer #3 (Remarks to the Author):

The authors provide a strong experimental validation combining NMR and MD simulation data to previously asserted binding of a rather dynamic RRM1 of SRSF1 to C nucleotides containing ssRNA through a canonical β -sheet surface (and indicates the minimal motif to be CN). They also have extended their observations to RRM1 of four other two-RRM-containing SR proteins namely SRSF4, SRSF5, SRSF6, and SRSF9 of humans and SRp55 of *Drosophila*. They further highlight the possibility of two distinct interaction specificity of RRM1 and RRM2 (joined with a flexible linker) to engage in a bimodal interaction and thus providing a complex variability. They also use this discovery to identify the mechanistic reason for the well-characterized phenomenon that SRSF1 cannot interact with the ESE1 of SMN2 exon 7. This absence of interaction, caused by a C-to-U transition in the exon 7 of SMN2 in comparison to that of SMN1, causes skipping of exon 7 of SMN2. Before moving onto the solution structure of

RRM1, they identified the residues on RRM1 that cause aggregation of SRSF1-RRM1. The solution structure aids in the identification of the amino acid residues of RRM1 involved in the interaction with RNA. Based on this information, the authors introduced mutations in RRM1 of full-length SRSF1 and showed that this mutant is capable of restoring inclusion of exon 2 of SMN2 in vivo. They also proved their rationale for this mutation in vitro by showing that this mutation enhances the interaction of the protein to 'U'-rich sequences in addition to 'C'-rich sequences. Then they carried out an ingenious SELEX experiment to establish the bimodal interaction of SRSF1 involving the affinity for CN of the RRM1 and for GGN of the RRM2.

This work is timely and has been long overdue as the authors have pointed out. Most importantly, this work is to be credited for its contribution to basic understanding of the mechanism of action of one of the most studied and fundamental group of splicing factors – the SR proteins. The extension of the observation to five SR proteins is notable. However, I believe, the presentation of the manuscript should be more focused and should attempt to discuss some of the other aspects of SRSF1 that have recently been unraveled.

Thank you for your positive feedback on our work and your comments that we took into account for the revision of this manuscript. To answer to your comments, we performed several additional experiments: new NMR titrations, ITC measurements with 8 different RNAs and a new bioinformatic analysis. All modifications performed in the text are in red. Please find our answer below each of your comment.

Major points:

Can it be distinguished (while assessing the importance of the nucleotide at the second position by titrating SRSF1 RRM1 with NNCGNN, NNCCNN, NNCTNN or NNCANN) whether the effect is not coming from flanking C nucleotides? Perhaps should have been compared with N=A, G and T.

We used degenerated nucleotides to not introduce any bias to the method and we think that the fact that the cytosines were randomly distributed on each side of the selected nucleotide should actually prevent this effect. In addition, the C_AG motif that we found to be specifically recognized by RRM1 is in agreement with the SELEX consensus sequence that was obtained in the Manley's lab from fully degenerated sequences (the consensus sequence was ACGCGCA, Tacke and Manley 1995 EMBO J.).

Is there any data suggesting both RRMs are binding simultaneously to GGA containing SELEX RNA to justify their argument of a flexible linker between 2 RRM motifs? Binding affinity data for sequences 5'-UCAUUGGAU-3' and 5'-UGGAUUUUUCAU-3' containing the CA motif are missing.

Yes, we know that the two RRMs are binding simultaneously to RNA as saturation is reached at a 1:1 protein:RNA ratio during our NMR titrations with the UCAUUGGAU and UGGAUUUUUCAU RNAs. This is now clearly mentioned in the text. We performed ITC titrations with these two RNAs and SRSF1 RRM12 and obtained K_d values of 58 nM and 55nM, respectively. These data are now shown in the new figure S7A.

In addition to the referee request, we performed ITC titrations of the SRSF1 RRM1 with AACAAA and AACGAA RNAs and obtained similar K_d values (21 μM and 25μM, respectively) (Fig. S4C) and tested the binding of the protein to CCCCCC and UUUUUU RNAs. As expected from our NMR analysis, we observed a binding only with the polyC RNA. The affinity is slightly higher than with the AACAAA and AACGAA RNAs (K_d of 11μM instead of 21-25μM), which is most likely due to an avidity effect (Fig. S4C). We also mentioned in the text the RNA-binding cooperativity observed when both RRMs are linked by their natural linker (K_d around 50nM instead of 20 μM and 0.7μM with separated domains).

In general, binding affinity analysis of RRM1/2 with a select set of RNA with both RRM binding sites will make the manuscript lot stronger.

We also tested the interaction of SRSF1 RRM12 with RNA mutants in which the CA motif was mutated to UU and obtained a decrease in affinity for both RNAs indicating clearly a contribution of the RRM1 in the binding (K_d values of 164 nM and 143 nM with UUUUUGGAU and UGGAUUUUUUUU RNAs, respectively) (Fig. S7A). Finally, we prepared a figure with an NMR overlay of SRSF1 RRM12 bound to UCAUUGGA, UGGAUUUUUCAU and UCAUUGGAUUUUUCAU (containing a CA motif on both sides of the GGA) RNAs (Fig S7B). Saturation was obtained at a 1:1 ratio for all RNAs and all the chemical shifts observed at this ratio with the UCAUUGGAUUUUUCAU RNA were located in between those observed with UCAUUGGA and UGGAUUUUUCAU RNAs (Fig. S7B). This result suggests the presence of an exchange between the two populations or at least no preference for one of the two binding modes, which is in good agreement with the similar affinity observed above for the two binding registers.

Can it be separated whether the RNA is binding to both at the same time or in succession by NMR experiment? Overlay of 1H-15N HSQC spectra measured with SRSF1 RRM1+2 YS free form and in the presence of UCAUUGGAU or UGGAUUUUUCAU RNA..

Only one binding event is observed with ITC in the presence of the two binding sites (Fig. S7A). In addition, when we look carefully at the titration performed with SRSF1 RRM12 and either UCAUUGGAU or UGGAUUUUUCAU RNA, all peaks from RRM1 and RRM2 experienced some chemical shift perturbations already at a 0.3:1 RNA:protein ratio. This indicates that both RRMs interact simultaneously, which is in good agreement with the cooperative binding that we observe with ITC (K_d of about 55nM with both RRMs instead of 600-700nM and 20μM for RRM2 and RRM1 alone, respectively).

The re-analysis of mouse and human (K562 and HepG2) CLIP data needs to be analyzed to check presence and distribution of C in more detail to strengthen the utility of this manuscript.

We have re-analyzed available CLIP data similarly to the SELEX data and the new Suppl. Fig. S8 shows the results. These new data also show some evidence of C enrichment, although the patterns are not easily interpretable, as they are not fully replicated even among CLIP data obtained from different cell lines. We modified the text according to the outcome of this new analysis.

The authors have identified two aspects of SRSF1 biochemistry – its interaction with RNA and its self-aggregation. However, they have not fully utilized this information to put forward models to explain known functions and behaviors of SRSF1. A recent publication (Nucleic Acids Res 2020 Jun 19;48(11):6294-6309) suggests that SRSF1 binds cooperatively to the pre-mRNA depending on availability of single-stranded sequence immediately upstream of the 5'SS, which then structurally remodels the pre-mRNA. This cooperativity could be explained by the interaction the authors have identified. Optionally the authors might want to use their Y37/Y72 mutants to examine if the cooperativity of the full-length proteins is caused by these residues. Additionally, the authors should invoke a discussion how self-aggregation and bimodal interaction help SRSF1 in remodeling the pre-mRNA.

The authors must include the RS domain of SR proteins in their discussion. RS domains are phosphorylated for activating SR proteins. It is also known to bind the branchsite. Although RS domain is not essential for splicing of several model pre-mRNA substrates tested in vitro, its presence in vivo warrants its inclusion while discussing sequence specificity of SRSF1 binding, particularly when the binding consensus is so degenerate.

We agree with the referee and included a part on SRSF1 ability to remodel RNA and the possible involvement of the RS domain in SRSF1 recruitment on RNA. However, we think that there is not enough experimental evidence to conclude about a clear binding with cooperativity of multiple SRSF1 proteins to RNA (no clear affinity measurement was shown and the term cooperativity or multimerization was not used in the manuscript mentioned by the referee). This recent publication showed that several molecules of SRSF1 could indeed interact with the β -globin pre-mRNA, but they could contact RNA independently from each other. In addition, the aggregation that we observe with the tyr37 and tyr72 of RRM1 is happening at high protein concentrations (required for NMR measurements) and may not occur in physiological conditions. Therefore, based on the new ITC measurements, we now mention in the text the RNA-binding cooperativity observed with the two RRMs of SRSF1, but did not extend the discussion to the inter-molecular RNA-binding cooperativity.

Minor Points:

The author observed the limitation of their mutation in rectifying the splicing of SMN2 gene (inclusion of exon 7 is accompanied by retention of intron 7 – Figure 3C). Yet, they advanced the idea of this mutation as a potential cure for SMA. This SRSF1 mutant being able to rescue skipping of exon 7 is a sufficient advancement of splicing biochemistry. This does not have to be (weakly) established as a cure for SMA. Therefore, the claim that this mutation could be a potential cure should be softened or possibly removed altogether from the manuscript.

We softened this claim in the text.

Too many glamor words- unexpected, bimodal/pendular, conserved non-conventional etc. The work is informative but bimodal interaction is not exactly novel. Neither is the recognition of C by RRM1.

It has never been suggested or shown in any other paper that the RRM1 of SRSF1 could have the ability to interact on each side of the RRM2 binding site. Therefore, we maintain that this mode of interaction was unexpected and should be underlined in the text. We removed the

“pendular” and additional glamor words from the text, but kept the “bimodal” word as it is clearly describing the mode of RNA recognition of SRSF1.

The fact that RRM1 binds to a distinct motif was known/envisioned but has not been investigated, it does not necessarily explain the difficulty to derive a consensus binding sequence for this protein so far.

We agree with the referee that it does not necessary explain the difficulty to derive a consensus binding sequence and modified the text.

-Figure 1A schematic needs better representation

A lack of clear definition of the interaction since restraints not sufficient to describe the RNA binding interface precisely and MD simulation was used.

We do not see what could be improved in the schematic representation that was used in the figure 1A. We mentioned the border of each RRM and the secondary structure of RRM1 based on our structures. The numbering is according to the pdb.

4E model is far-fetched.

The E panel was removed from the figure.

REVIEWERS' COMMENTS

Reviewer #1 (Remarks to the Author):

The authors have satisfactorily addressed by requests and erased all my concerns. I congratulate them on this excellent work.

Reviewer #2 (Remarks to the Author):

The authors appropriately addressed my minor comments. This is an important and timely contribution.

Reviewer #3 (Remarks to the Author):

Authors have appropriately responded to my comments. I recommend its publication.